# Visual Feedback and Virtual Reality in Gait Rehabilitation of Hemiparetic Children and Teenagers after Acquired Brain Injury: A Pilot Study

**DOI:** 10.3390/children9111760

**Published:** 2022-11-16

**Authors:** Daniele Panzeri, Chiara Genova, Geraldina Poggi, Sandra Strazzer, Emilia Biffi

**Affiliations:** Scientific Institute, IRCCS Eugenio Medea, Bosisio Parini, 23842 Lecco, Italy

**Keywords:** hemiparesis, virtual reality, rehabilitation, gait, children, adolescents

## Abstract

Hemiparesis as a term refers to a neurological disorder that can be extremely variable, especially with regard to walking abilities. Few works have assessed the use of virtual reality and biofeedback in children and adolescents with hemiparesis. The aim of this study is to provide insights about the rehabilitation of hemiparetic children and teenagers with visual biofeedback in a virtual reality environment. Six hemiparetic subjects (mean age 13.13 years, age range (7–18), 4 males) received 20 personalized rehabilitation GRAIL (Gait Real-time Analysis Interactive Lab) sessions plus 20 sessions of traditional physiotherapy. After an initial evaluation of rehabilitation needs, training focused on gait pattern correction (GP), walking endurance (WE), or gross motor functions (GMFs). All subjects were assessed for their gait analysis by GRAIL, the Gross Motor Function Measure (GMFM), and the 6–Minute Walking Test (6MWT) before and after rehabilitation. All subjects reached their rehabilitation goals, save for one who showed reduced collaboration. In addition, 4 subjects reached a better GP, 3 subjects reported improvements in WE, and 2 subjects improved in GMF. This personalized training with visual biofeedback delivered in a VR setting appears to be effective in modifying motor control and improving gait pattern, in addition to resistance and functional activities, in subjects with hemiparesis.

## 1. Introduction

In recent decades, the use of virtual reality (VR) and biofeedback in neurorehabilitation has been considered to be quite promising [1], but up to now, results are still controversial [2]. While some reviews point to some efficacy [3,4], others do not attribute it a greater effectiveness versus traditional therapy [5].

One of the main advantages of VR-based systems in rehabilitation is visual biofeedback: it helps subjects achieve a better understanding of their movements and greater control [6]. It also offers engaging environments and applications, encouraging patients’ compliance.

Many studies concerning the rehabilitation of hemiparetic patients by VR were conducted in adult populations (for reviews, see Zhang et al. [7] and Luque-Moreno et al. [8]). These systematic reviews reported the effectiveness of VR-based interventions on lower-limb functions in adults. Only few articles have addressed children or adolescents [9,10], with promising results.

However, demonstrating the efficacy of VR in modifying walking abilities in hemiparetic children and teenagers after acquired brain injury (ABI) poses two main challenges.

First, VR as a term refers to many different devices, from commercial consoles to sophisticated systems enabling a customized rehabilitation path [5]. This can lead to very different kinds of activities and, consequently, different outcomes. Collating data from heterogeneous devices makes it difficult to detect a clear trend in results.

Second, hemiparesis refers to a neurological disorder that can be extremely variable, especially with regard to walking abilities. This variability was reported by a number of studies that identified patterns of movement in hemiparetic subjects [11] and developed classification systems to group them, based on kinematic data [12], walking speed [13], foot contact [14], or electromyography data [15].

In addition to this, different patterns can be observed even within the same group [16], meaning that rehabilitation needs may be different, too.

This study was aimed at evaluating how a personalized program using visual biofeedback in a VR setting can impact on gait abilities of subjects with ABI-related hemiparesis.

## 2. Materials and Methods

### 2.1. Study Design

All the subjects recruited received an initial assessment (T0) to define their personalized rehabilitation path. For all the patients, the rehabilitation protocol consisted of 20 sessions, each lasting 45 min, with the GRAIL system and 20 sessions of traditional physiotherapy (TP), each lasting 45 min, over one month. At the end of the protocol (T1), T0 assessment was repeated.

### 2.2. Grail

The GRAIL (Gait Real-time Analysis Interactive Lab, Figure 1) is a laboratory suitable for motion analysis and VR-based rehabilitation consisting of a treadmill provided with two belts that can be driven at different speeds and equipped with integrated force platforms (16 channels, sample frequency 1000 Hz). The motion platform allows movements of antero-posterior pitch and lateral sway. An immersive semicircular screen is placed in front of the treadmill and projects VR environments, synchronized with the treadmill movement. For patients’ maximum safety, the system is also equipped with two handrails and a harness. The system uses a Vicon motion capture system (10 optoelectronic cameras, sample frequency 100 Hz) and 3 video cameras to perform motion detection and gait analysis. The integration of all these features provides multi-sensorial feedbacks (visual, proprioceptive, and auditive) during training.

### 2.3. Outcome Measures

#### 2.3.1. Gross Motor Function Measure

The GMFM is an 88-item scale divided into 5 sections: A—lying and rolling; B—sitting; C—crawling and kneeling; D—standing; E—walking, running, and jumping. All items are scored from 0 (most severe impairment) to 3 (healthy subjects), for a total of 264 points [17]. The total score is then divided by the maximum score in each dimension and reported as a percentage. The reliability of GMFM in patients with ABI has been reported in the literature [14].

#### 2.3.2. 6–Minute Walking Test

The 6MWT is used to evaluate walking endurance [18]. It measures the walking distance covered in six minutes at an affordable speed for each subject along a 25 m standardized path. The validity of the 6MWT in patients with ABI has been reported in the literature [19].

#### 2.3.3. Three-Dimensional Gait Analysis (3dga) over the Grail System

Multistep 3DGA was performed with the GRAIL system enabling motion data recording and processing in real time by means of the Human Body Model I (HBM-I). Accordingly, 25 markers were placed on specific anatomical landmarks of the body. Patients wore underwear during gait acquisition. After a ten-minute familiarization phase, about 40 steps were acquired. During 3DGA, no biofeedback was provided with the GRAIL.

### 2.4. Participants

Inclusion criteria were: children and teenagers with ABI-related hemiparesis. All the subjects had to be able to walk autonomously, without any device. 

Exclusion criteria were: age < 5 years, severe lower-limb spasticity affecting walking, recent lower-limb surgery, QI < 60, and behavioral problems (e.g., aggressiveness or anxiety) that would limit participation.

The protocol was approved by the ethical committee of Scientific Institute Medea (prot. 585, accepted on 20 June 2018) and conducted in accordance with the Declaration of Helsinki. Patients or their parents provided written informed consent.

### 2.5. Initial Assessment and Rehabilitation Program

Participants received a personalized program (addressed both with GRAIL and in TP) based on their initial assessment results, aimed at improving the most compromised aspects. To better group the identified needs, they were labeled as follows: gross motor functions (GMFs), walking endurance (WE), and gait pattern (GP).

A GMF rehabilitation goal was assigned when the total GMFM-88 score was less than 90%. 

The WE rehabilitation goal was assigned when the difference between the patients’ 6MWT result and the distance covered by age-matched healthy subjects (and normalized [20]) was greater than 30%. These cutoffs were defined arbitrarily, based on our clinical experience with other children and teenagers presenting hemiparesis.

As mentioned in the Introduction session, hemiparesis is characterized by many different walking patterns, and each one can have a different impact on daily life activities. Therefore, each patient was assessed for their gait pattern and assigned to a GP rehabilitation goal if their gait disturbance could be a current limitation or potentially cause future musculoskeletal disorders.

### 2.6. Training

Rehabilitation activities and exergames were grouped by the same 3 categories used for rehabilitation (i.e., GMF, WE, and GP), in order to assess the time each patient spent on each specific goal, by expressing the training time as a percentage of the total time trained. An example of activities is shown in Figure 2.

During the GRAIL training, WE was trained with exergames aimed to increase distance covered (e.g., walking long distances in a wood scenario) and improve spatiotemporal characteristics (e.g., modulate stride length to step over tiles). The GMF goal entailed activities of body weight transfer (e.g., performing a slalom while skiing), single-limb support activities (e.g., keeping one foot lifted to let cars pass below), dynamic balance (e.g., games where external perturbations were addressed to the patient), and multitasking activities (e.g., walking while trying to hit a target appearing on screen). The GP goal dealt with selective control exercises (e.g., directing a ball over a bar by foot movement) and gait pattern activities (e.g., activities during which kinematics and reference values were shown so that subjects could correct movement).

All the subjects were trained for all three categories, but training varied according to their rehabilitation need(s). In the TP, the subjects trained considering the same goals, trying to strengthen what they had learnt using the GRAIL and transfer them in the daily life activities. To increase WE, patients were asked to cover a gradually increasing distance training at different speeds or over slopes. The GMF goal was trained by exercising with body weight shifts (particularly over the impaired lower limb, which is usually characterized by reduced loading time), balance exercises (in static and dynamic conditions), and with everyday activities, such as climbing stairs and all posture changes. Finally, the GP goal was trained using exercises specific for the change in pattern shown by each subject.

## 3. Results

Six patients affected by hemiparesis as a consequence of acquired brain injury were recruited. Demographic data are reported in Table 1.

Initial assessment data are reported in Table 2. Table 3 shows the training time for each patient in the three specific domains (GP, GMF, and WE), expressed as percentage.

### Participants and Training Activities

S1 is a left hemiparetic child without relevant limitations in WE or GMF (see Table 2). His main impairment was the gait pattern. He showed an evident left drop foot, owing to a complete hypoactivation of all foot and ankle muscles, and foot contact occurred with the forefoot. As a consequence, he made compensatory knee and hip movements (see Figure 3, panel S1, T0). Our goal was to improve control of his left foot. During the first sessions of GRAIL, he carried out an exercise based on visual feedback, consisting of a vertical bar, over which the patient could move a ball, representing the movement of his left foot (see Figure 2F). This exercise was geared to improve his ability to recruit dorsiflexor muscles. In the following days, we tried to maintain this activity during gait exercises. As S1 quickly learned how to use visual biofeedback, we asked him to perform the “MM Gait”, an exergame in which graphs related to his kinematics were projected in real-time on screen so that he could compare his pattern against normative kinematics (see Figure 2E). We started showing one district at a time, beginning from the left foot followed by graphs of the knee and hip districts, gradually asking him to control all the segments simultaneously. During TP, the same goal was set, and he trained in activities enabling better recruitment of dorsiflexor muscles and their activation during gait.

At the end of the rehabilitative period, S1 could maintain the newly corrected pattern, even when no visual feedback was provided (Figure 3, panel S1, T1). Compared to kinematics at T0, when he showed a completely irregular foot pattern impacting on the knee and hip, kinematics at T1 was more similar to normative data for the ankle district and the knee and hip joints as well. The graphs show the increase in dorsiflexion during the swing phase.

S2 is a right hemiparetic girl showing moderate reduction (−26%) in walking endurance as compared to age-matched healthy subjects. On the other hand, she showed some balance difficulties and limitations in everyday life, as suggested by total GMFM scores of 86% (see Table 2). For this reason, activities mostly dealt with body weight transfer and balance, both in static and dynamic conditions (for a cumulative time of 49%, see Table 3) and, to a lesser degree, part pattern correction (28%) and walking endurance (24%).

To improve her body weight transfer ability using the lateral displacement of the body mass, she played games such as “Skiing” (see Figure 2B, in which she had to move her weight to the right and to the left to slalom between snowmans) or “Sailing a boat” (guiding with her body movements a virtual boat over a route). Her capacity to react to unexpected environmental stimulation was trained by exergames such as “Rope Bridge”, in which the subject walked over a swinging bridge (the treadmill pitched correspondingly) while trying to avoid seagulls coming from the front. During TP, S2 performed mostly balance activities, both standing and walking, over uneven ground, as well as postural passages and walking in outdoor setting, where endurance and balance were trained together.

At the final assessment, an increase in GMFM scores (+7% DIM D, + 10% DIM E, +6% total score) as well as a clinically relevant increase [21] in 6MWT (Table 4) were observed.

S3 is a right hemiparetic child. His GP was mainly impaired: he presented an overall reduction in right knee movement and his ankle pattern was characterized by marked dorsiflexion, during all gait cycles, and even more marked during the stance and pre-swing (see Figure 3). His initial rehabilitation plan consisted of activities with the help of visual biofeedback (similarly to S1), to help him increase his degree of movement at the knee and correct his ankle push-off. Unluckily, this kind of exercise that required much attention was too demanding for the patient. Therefore, he was switched to more playful activities to ensure his compliance. We focused on improving single limb support with games such as “Traffic Jam” (see Figure 2A), where the subject had to lift one foot to let some cars pass under his feet. Other exergames were geared to increase WE and his ability to modify spatiotemporal characteristics of gait: in “Microbes” (see Figure 2D), the subject could walk up to 10 min—speed was gradually increased by the therapist—interacting with the VR on screen. At every level, he had to perform different tasks so that he had to increase or reduce the length of his steps correspondingly, or to move forward and backward to collect the objects shown.

S3 showed a lack of compliance also during TP; therefore, all exercises (such as walking with obstacles and climbing over steps) were playful or sport-related activities. At the end of the project, S3 spent 65% of his GRAIL time in training to improve GMF functions (see Table 3) and showed a clinically significant improvement (+5%) on DIM D of GMFM.

S4 is a left hemiparetic boy presenting high scores on both 6MWT and GMFM. His main impairment was GP and consisted in a difficulty to control knee hyperextension during the stance phase. We trained him with the same visual biofeedback (the bar and the moving ball, see Figure 2F) used with S1, but, in this specific case, we worked on flexion-extension of the left knee. Initially, he had to play standing still. We designed the exercise so that when his knee moved toward hyperextension, corresponding to the inferior limit, an explosion feedback was provided. When S4 was able to perform this exercise, we asked him to keep his healthy foot over a stool so that his paretic limb carried most of his weight and he had to learn to control knee movement in full loading. Finally, we proposed the “MM Gait” activity (previously described), to give him the visual feedback of kinematic graphs during walking. In this exercise, S4 learnt to compare his left limb movement with the normative dataset and maintain correction during walking. During TP, he performed exercises to increase the recruitment of quadriceps and the gluteus and to stretch plantiflexors muscles. The therapist helped S4 to replicate the same activities in the real world: at the beginning, S4 was asked to avoid hyperextension in standing conditions and then gradually to perform it even during walking. Figure 3 shows a comparison between T0 and T1 kinematics: S4 modified his pattern, reducing the pre-rehab knee hyperextension and ankle plantiflexion.

S5 is a left hemiparetic girl. She presented with the longest time from injury (74 months). WE was one of her main rehabilitative needs as she showed a reduction of 38% in 6MWT as compared to normative data reported by Geiger and colleagues [20] (see Table 2). Moreover, her GP was characterized by a reduction in impaired knee movements, both in flexion during the swing phase, and in extension during the stance. A reduction in antigravity movements was also seen at the ankle level, which was maintained dorsiflexed during the whole gait cycle, especially during the stance. To compensate for this limitation, she kept even the non-impaired knee flexed during the stance (see Figure 3, panel S5, T0). In light of this, we alternated activities to increase WE and others based on kinematic biofeedback in order to improve the paretic limb kinematics during the stance phase and avoid compensatory movements of the non-affected lower limb.

She spent 22% of the training time to increase walking speed, endurance, and spatiotemporal characteristics with exergames such as “Microbes” (previously described), “A walk over the board” (a game simulating a walk along the dock, in self-paced modality, during which she had to increase or decrease her speed in order to make objects and animals appear by the sea) and “Snow memory” (a game in which she had to modulate her walking cadence and step length to move a cursor over memory cards and unveil them).

To improve her pattern of movement (40% of GRAIL time), she played, for instance, “Adaptability”, an exergame in which she had to lift her knee while walking to reach the target on screen, with the goal to increase knee or hip flexion. To improve the stance phase of the non-impaired limb, she played “Waves”, a game developed by our group during which she had to stand over a boat rolling over the waves, trying to keep her balance; simultaneously, the game allows one to activate only one belt to stimulate single limb support. In this situation, S5 had to carry her weight with the right (non-impaired) limb, while the left limb was trailing forward and backward. In this way, it was possible to simulate the stance phase of the right lower limb, during which S5 showed a flexion attitude at the knee, and she was asked to try to maintain her knee extended. She trained PT using the same activities, so she alternated the long walking path and exercises with obstacles and steps, with the aim to increase active movement of her left knee.

Assessment at T1 (Table 4) shows clinically relevant improvements in 6MWT (+49%). Moreover, S5 learnt how to reduce compensatory movements of the non-affected side, and correctly perform hip and knee extension. Finally, she acquired a smoother control on the impaired knee and increased her impaired ankle activity during pre-swing (Figure 3, panel S5).

S6 is a right hemiparetic boy. He presented with the shortest time since the event (1 month). All three aspects of WE, GP, and GMF were considered as training goals as he presented with significant limitations in all of them (see Table 2). Specifically, he showed a marked limitation in walking resistance (−68%) and his gait pattern showed an overall decrease in activity on the affected limb. The GMFM-88 at T0 was 78%. In the first part of the training, we aimed at helping him get used to the treadmill stimulation and gradually increase his walking speed using exergames such as “Walk on the wall of the castle” (it simulates a walk on the medieval walls of the city, without any specific request). Then, we trained his capacity to move his body weight over the impaired limb with games such as the previously described “Skiing” (see Figure 2B) or “Sailing a boat”. When he improved these movements, we switched to activities such as “Forest road” (he had to lift the hands away from the support and use them to hit birds and butterflies passing-by while walking). Finally, he used “MM Gait” (see Figure 2E) with the kinematic graphs to improve his right limb kinematics during the stance phase. During PT, he trained the same goals: he had to stand with his weight over the impaired limb, while the left limb was located over a stool. In this way, he could gradually increase the stance time over the impaired limb and the muscular strength. He also performed walking activities without support, gradually increasing the distances covered and the difficulty of the path.

As S6 had to train all the three goals, the training time was spread, even if mainly dedicated to GMF activities. As a result, he reported clinically significant improvements in DIM D (+10%), DIM E (+30%), and total score (+14%) of GMFM. For what concerns WE, S6 increased the distance travelled at T1 of 151.5 m. Regarding GP, he reduced the adduction and endo-rotation of the impaired hip and corrected his paretic knee and ankle pattern (see Figure 3, panel S6). Additionally, while at T0, he needed to use hand support for his safety, and at T1, he learnt to walk without any support.

## 4. Discussion

In recent years, VR application in the rehabilitation of hemiparetic subjects proved effective in improving walking characteristics such as speed, distance, and endurance [22]. By contrast, gait pattern change seems to be quite challenging, and up to now, there has been little encouraging evidence [6,10,23].

Booth et al. [6] showed how training delivered through visual biofeedback in VR to children affected by cerebral palsy was effective in inducing a modification in their motor activities. Our participants suffered an ABI and were trained for 20 sessions. During this period, they learnt how to modify their muscular activity and correct their gait pattern. Repetitive training in combination with traditional physical therapy appears to be successful in improving gait pattern, not only during the training but also at the final assessment without visual feedback.

Reviews or randomized clinical trial articles obviously present a more rigorous design, but in our study, we wanted to describe how a customized VR-based program can be delivered and how this can impact on each single subject. To do this, every patient was assessed, and a rehabilitation goal was defined and tailored according to each subject’s needs. These needs were different for the 6 subjects recruited, so activities to improve WE, GMF, or GP were differently targeted.

Interestingly, we observed a change for every goal we set.

WE was set as a rehabilitation goal for two subjects (S5 and S6). These patients and S2 spent more than 20% of their training in activities to increase walking endurance and spatiotemporal characteristics. They all showed a clinically significant improvement on the 6MWT [21]. On the other hand, three subjects (S1, S3, and S4) showed a worsening score on the 6MWT, and in two cases (S3 and S4), this reduction was clinically significant. Training time in WE activities for these three subjects was less than 20%. Moreover, S1 and S4 focused on correcting their GP, and it may be likely that they reduced their walking speed in order to maintain the kinematic corrections during walking.

The two subjects (S2 and S6) with the GMF goal trained consistently and reported clinically significant improvements [21] on the total GMFM score, as well as on D and E dimensions. S3 also showed an improvement on the GMFM D dimension after switching to this GMF goal as he was poorly compliant with GP.

GP improvement was a goal for five subjects (S1, S3, S4, S5, and S6). Excluding S3, three of them, S1, S4, and S5, trained for more than 40% of their time and could effectively improve their GP.

The most relevant aspect of this work was the improvement in GP we could identify. We showed how a personalized VR training, through visual feedback, could lead them to modify their gait pattern. In fact, all subjects involved in this goal showed improvements, regardless of the time since the brain injury was sustained. 

This study had some limitations. First, the study had a limited number of patients and no group analysis was performed. Thus, the study has limited inference power and lacks generalizability. Second, our study lacked a follow-up assessment. It would be interesting to understand if these improvements were maintained over time. Another limitation was that we collected gait analysis data to detect the change in subjects’ activities. Further studies should be geared to measure increases in force or different activation timing, based on electromyography.

Nevertheless, some conclusions can be drawn. The first few months after injury are considered the time period during which most of the recovery takes place [24]. Accordingly, S6 (shortest time since injury) showed significant improvements in all the three areas (GP, WE, and GMF). However, S5 (74 months after injury) reported some clinically meaningful improvements, both in walking endurance and in gait pattern, too.

Subject participation appears to play a fundamental role in this personalized rehabilitation plan: S4 was very determined in improving his pattern and participated in activities and could learn a better control of his knee during gait. In contrast, S3 showed the least collaboration and the smallest improvement within the group.

To conclude, a personalized rehabilitation plan based on biofeedback in VR appears to be a useful approach for the rehabilitation of hemiparetic (pre)adolescents. The earlier it is applied, the better. However, it is worth implementing it, even after some time from the brain injury. 

## Figures and Tables

**Figure 1 children-09-01760-f001:**
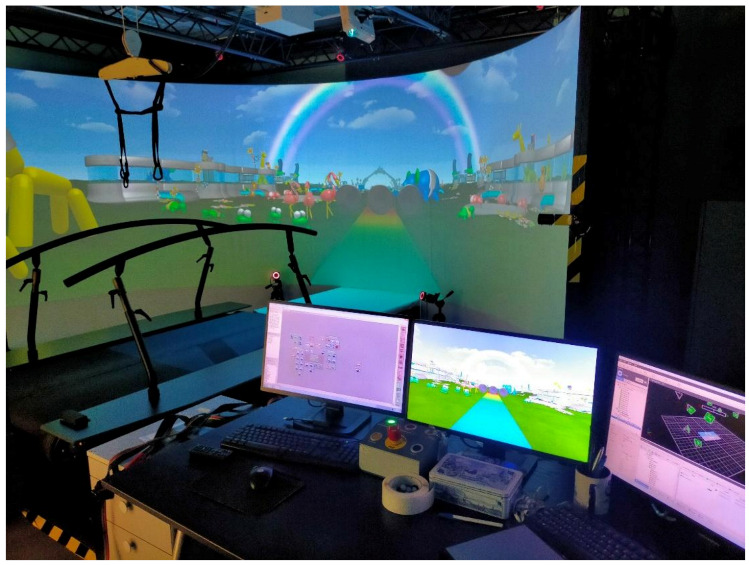
The GRAIL.

**Figure 2 children-09-01760-f002:**
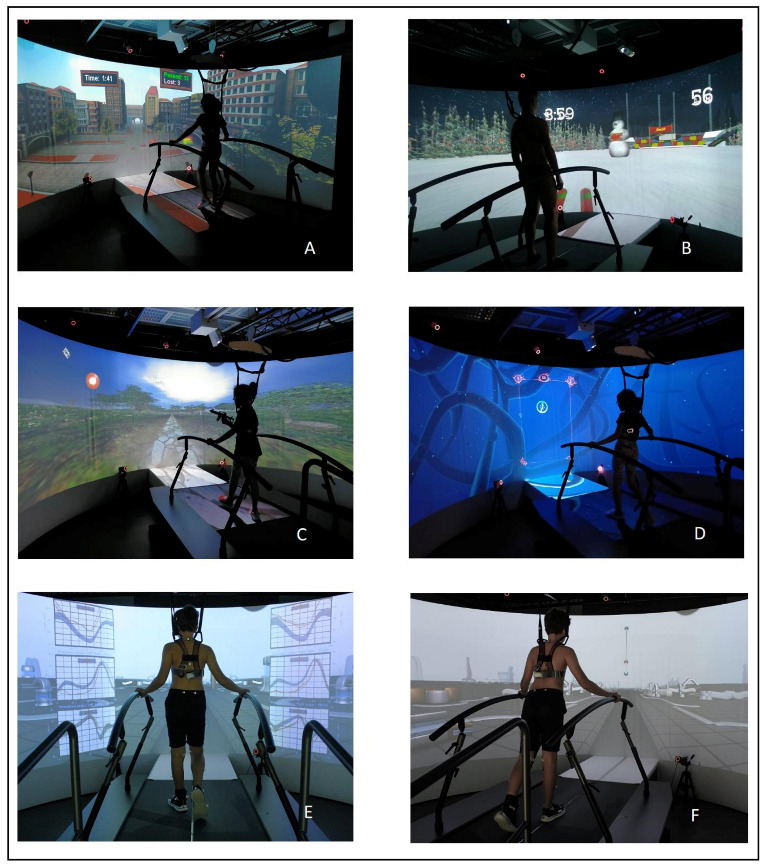
Examples of GRAIL activities: (**A**) (traffic jam) and (**B**) (ski) are GMF activities; (**C**) (hit the balloon) and (**D**) (microbes) are WE activities; (**E**) (MM gait) and (**F**) (Re-training) are GP activities.

**Figure 3 children-09-01760-f003:**
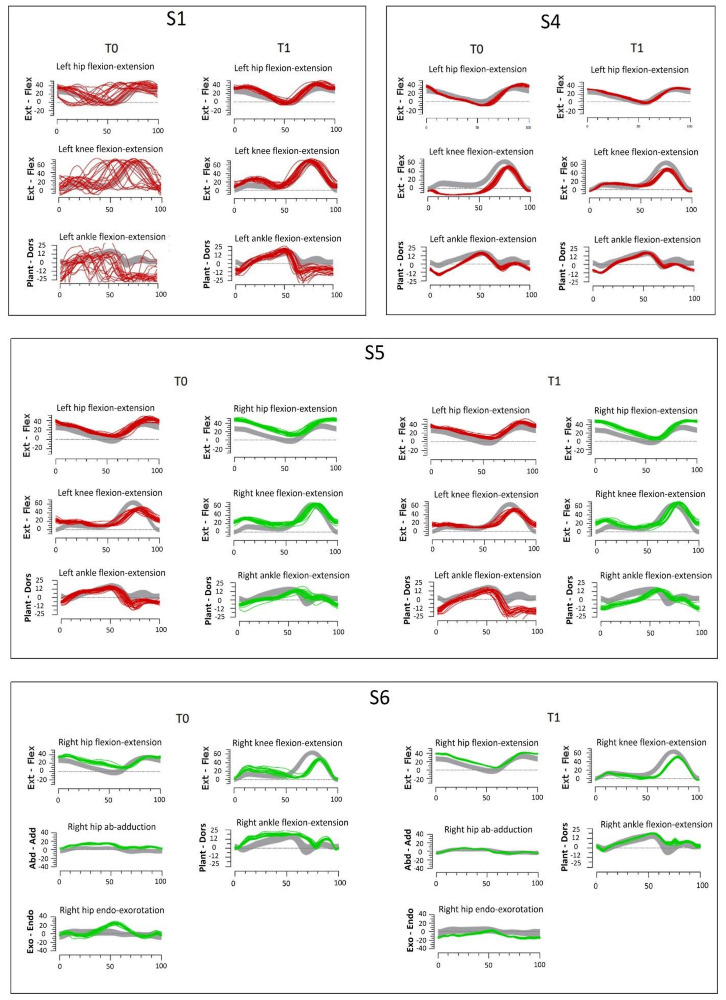
Kinematics of the four subjects that trained their gait pattern. Data from each subject are collected in a rectangle. For every patient, we collected data concerning kinematics aspects from T0 and T1. Red lines represent left side; green lines right side. For each district, we reported normative data of age-matched healthy subjects.

**Table 1 children-09-01760-t001:** Demographic data.

Subject ID	S1	S2	S3	S4	S5	S6
**Age (years)**	9.8	18.3	7.8	15.9	12.8	14.3
**Sex**	m	f	m	m	f	m
**Impaired side**	left	right	right	left	left	right
**Time from injury (months)**	34	63	11	17	74	1
**Etiology**	Resective Surgery for epilepsy	Stroke	Stroke	Traumatic brain injury	Brain tumor	Brain tumor

**Table 2 children-09-01760-t002:** T0 assessment for the six subjects. Rehabilitation need(s) are identified as GP = gait pattern; GMF = gross motor functions; WE = walking endurance. §: Values are reported as variation versus same-age healthy subjects (Geiger et al. [20]), expressed in percentage. * Regarding S3: initial assessment had pointed to GP as rehabilitation need but he mostly trained with playful games, which are considered GMF activities owing to poor collaboration.

	S1	S2	S3	S4	S5	S6
**6MWT (m)**	525	490.2	442.2	557.4	405.4	221.5
**6MWT %** ^§^	−21%	−26%	−24%	−20%	−38%	−68%
**GMFM% Dim D**	95	85	92	100	95	77
**GMFM% Dim E**	96	79	86	99	93	49
**GMFM-88%**	97	86	95	100	98	78
**Gait analysis**	see Figure 3					
**Rehabilitation needs**	GP	GMF	GP *	GP	WE, GP	WE, GP, GMF

**Table 3 children-09-01760-t003:** Training time for each specific goal, expressed as percentage of total GRAIL training. White cells represent training time lower than 20%; light gray cells represent training time between 20% and 35%; dark gray cells represent training time higher than 35%.

	S1	S2	S3	S4	S5	S6
Walking resistance and spatiotemporal characteristics (WE)	17	24	14	18	22	24
Body weight transfer, dynamic balance, single-limb support (GMF)	32	49	65	27	38	57
Selective motor control and kinematics (GP)	52	28	21	55	40	19

**Table 4 children-09-01760-t004:** Outcome measure at T1. Values are reported as difference from T0 assessment. Bold indicates clinically significant values [21].

	S1	S2	S3	S4	S5	S6
**6MWT (m)**	−22.6	**41.1**	**−34**	**−67.2**	**49.2**	**151.5**
**GMFM% Dim D**	2	**7**	**5**	0	0	**10**
**GMFM% Dim E**	4	**10**	2	0	1	**30**
**GMFM-88 %**	2	**6**	2	0	0	**14**

## Data Availability

The functional assessment presented in this study is available in Table 2. Gait data are openly available in Zenodo at LINK DOI 10.5281/zenodo.5862253.

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
