# Peer review of "Visual Feedback and Virtual Reality in Gait Rehabilitation of Hemiparetic Children and Teenagers after Acquired Brain Injury: A Pilot Study"

_children, 2022, doi:10.3390/children9111760_

Round 1
Reviewer 1 Report
The manuscript investigates the effect of Visual feedback and virtual reality in gait rehabilitation of hemiparetic children and teenagers with 6 childs through pre-post design. The manuscript is ambiguous and very difficult to follow and understand. It is poorly reported with a lot of English and reporting mistakes. Please find the following comments;
1- The study should firstly define the included participants and the inclusion criteria.
2- is the Gross Motor Function Measure a valid tool in assessment for the included different etiologies.
3- Revise the title to '' Visual feedback and virtual reality in gait rehabilitation of hemiparetic children and teenagers after acquired brain injury: a pilot study''.
4- Are the childs with acquired brain injury require special consideration in rehabilitation protocols compared with other pediatric population. Please justify the inclusion criteria.
Reviewer 2 Report
Thanks for the review opportunity.
It is thought that supplementation for the necessity of research is necessary. The mere fact that many studies have been conducted on adults and the lack of studies with children or adolescents is insufficient rationale to conduct research. In addition, specific suggestions should be made on what the main contents of the studies on adults were. It is necessary to conduct a review of previous studies related to the dependent variables of this study.
The reliability and validity of the measurement tools should be presented.
The criteria for exclusion are ambiguous. What is the criterion for Presence of severe spasticity? What were the specific examples of problem behavior?
In order to generalize the research results, it is necessary to conduct a nonparametric test on the pre- and post-test results of 6 subjects and to revise the results based on the analysis results.
In the case of discussion, it is expected that many parts will be modified according to the revision of the results.
Round 2
Reviewer 1 Report
The authors answered the comments. The only comments is regarding the reporting the results, they should be shorter and more informative.
Reviewer 2 Report
Thank you for revising the manuscript according to my comments.